

**Source, composition, and environmental implication of neutral**
**carbohydrates in sediment cores of subtropical reservoirs, South**
**China**
Dandan Duan[1,2], Dainan Zhang[1,2], Yu Yang[1], Jingfu Wang[3], Jing'an Chen[3], Yong Ran[1]
[1]State Key Laboratory of Organic Geochemistry, Guangzhou Institute of Geochemistry, Chinese Academy of Sciences,
Guangzhou, Guangdong 510640, China
[2]University of Chinese Academy of Sciences, Beijing 100049, China
[3]State Key Laboratory of Environmental Geochemistry, Institute of Geochemistry, Chinese Academy of Sciences, Guiyang
55002, China
Email: Dandan Duan dan113133@126.com; Dainan Zhang zhangdainan88@163.com; Yu Yang yangyu@gig.ac.cn; Jingfu
Wang wangjingfu@vip.skleg.ac.cn; Jing'an Chen chenjingan@vip.skleg.ac.cn
*Correspondence to:* Yong Ran (yran@gig.ac.cn)




**Abstract.**

Neutral carbohydrates along with algal organic matter (AOM) content, carbon isotopic composition, and elemental C/N ratios were investigated in three sediment cores of various trophic reservoirs in South China. Neutral monosaccharides, AOM, and carbon stable isotope ratios were determined by high-performance anion-exchange chromatography, Rock-Eval pyrolysis, and Finnigan Delta Plus XL mass spectrometry, respectively. The carbon isotopic compositions were corrected via the Suess effect. The concentrations of total neutral carbohydrates (TCHO) range from 0.51 to 6.4 mg/g at mesotrophic reservoirs, and from 0.83 to 2.56 mg/g at an oligotrophic reservoir. Monosaccharide compositions and diagnostic parameters indicate a predominant contribution of phytoplankton in each of the three cores, which is consistent with the results inferred by the corrected carbon isotopic composition and C/N ratios. The sedimentary neutral carbohydrates are largely structural polysaccharides and, thus, are resistant to degradation in the sediments. Moreover, the single neutral carbohydrate content is highly related with the carbon isotopic composition, algal productivity (hydrogen index), and increasing mean air temperature during the past 60 years. The nutrient input, however, is not a key factor affecting the primary productivity in the three reservoirs. The above evidence shows that neutral carbohydrates have been significantly elevated by climate change even at low latitude regions.

## 1 Introduction

Carbohydrates are the most abundant compounds in the biosphere, and they are present in the natural environment as both structural and storage compounds of aquatic and terrestrial organisms, comprising about 20–40 wt% of plankton (Parsons et al., 1984), more than 40 wt% of bacteria (Moers et al., 1993), and more than 75 wt% of vascular plants (Moers et al., 1993). Due to their high biological reactivity and availability, carbohydrates are preferentially utilized by heterotrophic organisms (e.g., bacteria and fungi) during transport of organic matter from water columns to underlying sediments (Hernes et al., 1996; Khodse et al., 2007), resulting in the preservation of more refractory structural carbohydrates in sedimenting particles (Cowie and Hedges, 1994; Burdige et al., 2000; Jensen et al., 2005; He et al., 2010). Moreover, the compositional signature of structural carbohydrates depends more on planktonic sources than the diagenetic pathway (Hernes et al., 1996). Thus, although carbohydrates exhibit different degrees of degradation, some structural fractions are selectively preserved and their compositions are mostly unchanged, which can be used as a powerful tool for elucidating sources, deposition processes, and diagenetic fates of organic matter (OM) in aquatic environments (Cowie and Hedges, 1984; Moers et al., 1990; Hicks et al., 1994, Meyers, 1997; Unger et al., 2005; Aufdenkampe et al., 2007; Skoog et al., 2008; Khodse and Bhosle, 2012; Panagiotopoulos et al., 2012).

Carbon isotope analyses in sedimentary OM offers an important tool for reconstructing the history of nutrient loading and eutrophication in lacustrine sediments (Schelske and Hodell, 1991; 1995). Phytoplankton preferentially removes dissolved $^{12}CO_2$ from epilimnetic water and depletes $^{12}C$ in the remaining dissolved inorganic carbon (Hodell and Schelske, 1998). As supplies of $^{12}CO_2$ become diminished, phytoplankton discriminates less against $^{13}C$ and sinking OM, incorporating more $^{13}CO_2$. Therefore, increased or decreased productivity can be reflected by enriched or depleted values of $\delta^{13}C$ in OM from the underlying sediments. However, during recent years, the $\delta^{13}C$ value in atmospheric $CO_2$, water column, and sedimentary OM have been significantly diminished and impacted by the Suess effect (Schelske and Hodell, 1995), which is defined as the change in the abundance of carbon isotopes ($^{14}C$, $^{13}C$, $^{12}C$) in natural OM reservoirs due to anthropogenic activities (e.g., fossil fuel combustion) (Keeling, 1979). Thus, the Suess effect needs to be considered when applying $\delta^{13}C$ in lacustrine sediments as a proxy for aquatic productivity. Although O'Reilly (2005) had not corrected for the Suess effect in the heterotrophic Lake Tanganyika in Africa, Verburg (2006) found that the corrected $\delta^{13}C$ values were used as a productivity proxy. In the Pearl River Delta, the development of industrialization and urbanization could enhance the Suess effect during recent years. Hence, it needs to correct for the Suess effect.



Several studies have shown that climate warming plays a significant role to algal productivity by using the S2 proxy in the Arctic lakes during recent decades (Outridge et al., 2005; Outridge et al., 2007; Stern et al., 2009; Carrie et al., 2010). However, Kirk (2011) investigated 14 Canadian Arctic and sub-Arctic lakes and found that the relationship between the S2 proxy and climate warming was irrelevant. In addition, only limited investigations have focused on the impact of global change on aquatic productivity in subtropical lakes (Hambright et al., 1994; Smol et al., 2005). Therefore, the above observations call for more investigations on the effects of trophic levels, early diagenesis, and sources of organic matter on the relationship between algal productivity and climate warming.

For this investigation, subtropical reservoirs in rural areas having different trophic states were chosen. Our purpose is to assess the source and diagenetic state of the carbohydrates, and their relationship with algal productivity in sediment cores by using carbohydrate compositions combined with Rock-Eval parameters, carbon isotopic composition, and elemental C/N ratios. In addition, trace metals data (Cu and Zn) cited from our previous paper (Duan et al., 2014) are used to help understanding the source of carbohydrates. Besides, neutral carbohydrates and recorded temperature data were statistically analyzed to explore the effects of climate warming on the historical variations of neutral carbohydrates in subtropical regions over the last four decades.

## 2 Materials and Methods

### 2.1 Study area and sample collection

Zengtang reservoir (ZT), Lian'an reservoir (LA), and Xinfengjiang reservoir (XFJ) with different depths and trophic states were chosen for this investigation. The detailed description of the study sites are shown in the previous literature (Duan et al., 2015). In brief, ZT is a shallow, polymictic reservoir with a mesotrophic level, whereas LA is a deep and mesotrophic reservoir. XFJ is a deep and oligotrophic reservoir. Both LA and XFJ are monomictic reservoirs. The most abundant algal species in the ZT, LA, and XFJ reservoirs are green algae, cyanophyta, and diatoms.

Undisturbed sediment cores were collected from the central part of the studied lakes using a 6 cm diameter gravity corer with a Plexiglass liner in 2010 and 2011. The water depths for the sampling sites at ZT, LA, and XFJ reservoirs are 3 m, 17 m, and 36 m, respectively. The core liners were put down slowly in order to avoid disturbance. The sediment cores were sliced into 2 cm thick intervals using extrusion equipment. It is noted that the top four slices of the ZT core were merged into two intervals (0–4 cm and 5–8 cm) due to the insufficient amount of sample for neutral sugar analysis. All subsamples were immediately placed in plastic bags, sealed, and stored at low temperature (0–10 ℃), and then were quickly transported to the laboratory, where they were freeze-dried and stored until further analysis.

### 2.2 Physicochemical properties in water

Vertical and temporal variation of chlorophyll a, dissolved oxygen, and temperature in the water column of the LA reservoir were recorded by a CTD-90M probe (Sea & Sun Technology, Germany) in increment mode, which enables us to carry out a great number of profile records in the field (Fig. S1 in the supporting data). For the XFJ reservoir, the physicochemical record was conducted only in March 2014. The lack of data from the ZT reservoir is due to its reconstruction after the sediment core sampling.

### 2.3 Stable carbon isotopic analysis

Samples were initially decarbonated by the moderate HCl solution, and then the stable carbon isotopic composition was measured by a Finnigan Delta Plus XL mass spectrometry. The $\delta^{13}$C values (‰) were given by the equation below:

$$\delta^{13}C (‰) = (R_{sample}/R_{standard} - 1) \times 1000$$




where R is the $^{13}$C/$^{12}$C ratio and the standard is the Pee Dee Belemnite. Black carbon (Product ID: GBW 04408) was used as
the reference standard for the determination of accuracy and precision. The precision of $\delta^{13}$C for the replicates was $<$
0.19‰ $\pm$ 0.12 (n = 40).

Measured values of $\delta^{13}$C were corrected for the Suess effect with the following polynomial equation (Schelske and

Hodell, 1995), where t is time (in yr):

$\delta^{13}$C (‰) $= -4577.8 + 7.3430t - 3.9213 \times 10^{-3}t^2 + 6.9812 \times 10^{-7}t^3$

The calculated time-dependent depletion in $\delta^{13}$C induced by fossil fuel combustion since 1840 was subtracted from the

measured $\delta^{13}$C for each dated sediment section.
**2.4   Neutral sugar analysis**
Sediment samples (about 5 mg) from the three reservoirs were weighted and hydrolyzed in glass ampules with 12 M
$H_2SO_4$ for 2 h at room temperature. After nine milliliters of Milli-UV + water were added (1.2 M $H_2SO_4$, final concentration
of acid), the ampules were flame-sealed and the samples were stirred and hydrolyzed in a 100 ℃ water bath for 3 h. The
hydrolysis was terminated by placing the ampules in an ice bath for 5 min. Then, the deoxyribose was added as the internal
standard (Michael et al., 2015). Before instrumental analysis, the samples were run through a mixed bed of anion (AG 2-X8,
20–50 mesh, Bio-Rad) and cation (AC 5OW-X8, 100–200 mesh, Bio-Rad) exchange resins (Michael et al., 2015).
Self-absorbed AG11 A8 resin was utilized to remove the acid. The volume of resin needed for complete neutralization
depended on the amount of acid used for hydrolysis (Michael et al., 2015). After purification and desalting with a mixture of
cation and anion exchange resins, neutral sugars were isocratically separated with 25 mM NaOH on a PA 1 column in a
Dionex 500 ion chromatography system, which was equipped with a pulsed amperiometric detector (PAD) detector (model
ED40) (Michael et al., 2015). The detector setting was based on Skoog and Benner (1997). Chromatographic data were
recorded with a personal computer equipped with Hewlett Packard Chemstation software (Skoog and Benner, 1997; Kaiser
and Benner, 2009).

For every ten analyses, a blank sample and a duplicate sample were analyzed to check accuracy and precision. No

neutral sugars were detected in the blanks. Only glucose, galactose, mannose, rhamnose, fucose, and xylose were detected
and analyzed in samples due to the loss of ribose in the process of acid hydrolysis. The recovery of the monosaccharides in
the sediments ranged from 73% to 95%. The analytical precision of duplicate samples performed on different days was
within $\pm$3% for glucose and $\pm$5% for other sugars. In this study, the total neutral carbohydrates (TCHO) are defined as the
sum of all identified monosaccharides.
**3   Results**
**3.1   Physicochemical properties of water**

As shown in Fig. S1 in the supporting data, chlorophyll a concentrations in the water column of LA are higher in spring

and summer than in fall and winter, which is consistent with the seasonal distribution of the content of dissolved oxygen.
The vertical profiles of chlorophyll a and dissolved oxygen also show similar patterns; depleted dissolved oxygen in the
hypolimnion accompanies low content of chlorophyll a throughout the year, suggesting the important role of oxygen content
in the growth of productivity. In general, the bottom sediments in LA are mostly under anaerobic conditions. Water
temperature is higher in summer and fall, resulting in thermal stratification in the water column. During winter, the lake
mixes from top to bottom due to the decrease in temperature. The nutrient can be brought to deeper depths and may increase
the productivity in the entire water column.





The concentration of chlorophyll a (average: 0.94 µg/L) in the water column of XFJ is much lower than that in LA. The
oxygen content varies from 9 mg/L at a depth of 1 m to 8.54 mg/L at a depth of 36 m (Fig. S1 in the supporting data),
suggesting the bottom sediments are under aerobic conditions.

### 3.2   Characteristics of OM

The data from Rock-Eval pyrolysis are listed in Table S1 in the supporting data, which were reported previously (Duan
et al., 2015). As shown in Table S1, pyrolytic parameter S1 and S2 represent the fractions of hydrocarbons (HC) released
during the pyrolysis step, where S3 was derived from the fractions of CO and $CO_2$ released during the pyrolysis and
oxidation procedures. The hydrogen index (HI) is calculated by normalizing the contents of S2 to TOC (Duan et al., 2015).
TOC and hydrogen index (HI) are in the ranges of 0.78–2.98 and 114–231 mg HC/g TOC, respectively, in the ZT core; in
the ranges of 0.88–4.31 and 151–229 mg HC/g TOC, respectively, in the LA core; and in the ranges of 0.47–1.76 and
141–196 mg HC/g TOC, respectively, in the XFJ core. In general, the HI values are enriched in the surface layers of all the
sediment cores.
$\delta^{13}C$ values (‰) range from −22.2‰ to −21.6‰ in the ZT core, from −26.4‰ to −24.1‰ in the LA core, and from
−27.2‰ to −23.1‰ in the XFJ core, with the average values of −21.9‰, −25.4‰, and −24.9‰, respectively (Fig. 1). After
the correction for the Suess effect, $\delta^{13}C$ values (‰) vary from −21.7‰ to −19.9‰ in the ZT core, from −5.1‰ to −23.9‰ in
the LA core, and from −25.5‰ to −22.9‰ in the XFJ core, with the average values of −20.8‰, −24.6‰, and −24.1‰,
respectively (Fig. 1).
Elemental C/N ratios vary from 3.51 to 9.34 in the ZT core, from 4.12 to 14.7 in the LA core, and from 2.3 to 10.1 in
the XFJ core, with the mean values of 5.56, 8.15 and 5.14, respectively (Fig. 1).
The detailed results from $^{210}Pb$ and $^{137}Cs$ radiometric dating was reported previously (Duan et al., 2015). The mass
accumulation rates (MAR) were cited and displayed in the Table S1 in the supporting data. As the ZT, LA, and XFJ
reservoirs had been small lakes before the dam construction, the chronological records for the ZT, LA, and XFJ cores are
longer than the time scales of the dam construction.

### 3.3   Neutral sugar data

The concentrations of seven monosaccharides (glucose, galactose, mannose, arabinose, rhamnose, fucose, and xylose)
are presented in Figure 2, and/or listed in Table S2 in the supporting data. Glucose (0.2–2.34 mg/g) is the most abundant
sugar in all the reservoir sediments, followed by galactose (0.09–1.2 mg/g), mannose (0.03–0.92 mg/g), and xylose
(0.01–0.95 mg/g). Concentrations of arabinose (0.05–0.77 mg/g) , rhamnose (0.04–0.67 mg/g), and fucose (0.02–0.43 mg/g)
are relatively low in the reservoir sediments. The TCHO concentrations at the ZT, LA, and XFJ reservoirs range from 1.94 to
5.36 mg/g, from 0.51 to 6.4 mg/g, and from 0.83 to 2.56 mg/g, respectively, and show decreasing downcore trends in all the
sediment cores. Carbohydrate yield (%) is the molar concentration of monosaccharide carbon normalized by TOC.
Carbohydrate yield (%) ranges from 7.08 to 10.9% in the ZT core, from 2.31 to 13.53% in the LA core, and from 1.93 to
12.52% in the XFJ core, with average values of 8.68%, 7.79%, and 7.33%, respectively (Fig. 3).
Compositions of neutral sugars in the three sediment cores are calculated based on their concentrations. Glucose
(21.1–27.6 mol%) is the most abundant monosaccharide, followed by mannose (14–21.5 mol%), galactose (14.7–18.7
mol%), arabinose (10.7–17.1 mol%), rhamnose (11–12.9 mol%), xylose (8.8–10.1 mol%), and fucose (3.2–7.2 mol%) in the
ZT sediments. Glucose (22.2–37.4 mol%) is also the most abundant monosaccharide, followed by galactose (12–20 mol%),
mannose (11.1–20.5 mol%), arabinose (10.7–14.4 mol%), rhamnose (6.9–11.9 mol%), xylose (5.09–18.32 mol%), and
fucose (3.3–7.74 mol%) in the LA sediments. For the XFJ core, glucose (18.5–48 mol%) is still the most abundant
monosaccharide, followed by galactose (12.3–3.7 mol%), mannose (3.8–23 mol%), arabinose (8.3–14.5 mol%), rhamnose
(6–11.2 mol%), xylose (0.75–21.6 mol%), and fucose (2.9–8.2 mol%) in the sediment core.





### 3.4 Meteorological records of the studied areas

The five-year moving average temperature ($T_5$) was calculated from the reported database (Duan et al., 2015). The mean air temperature in the Guangzhou area has increased by about 1.5 ℃ since 1960, and the mean air temperature in the Heyuan area has increased by about 1.52 ℃ between 1957 and 2004. Therefore, the above data suggests a significant trend in climate warming in the investigated areas during the last six decades (Duan et al., 2015). The annual hours of daylight in Guangzhou and Heyuan on the time scale of 60 years have been obtained from the China Meteorological Data Sharing Service System (CMDSSS). The annual hours of daylight in both areas are somehow variable and show a progressively decreasing trend from 1950 to 2010 (Fig. S5).

### 4 Discussion

#### 4.1 OM characteristics in sediment cores

The content and composition of sedimentary OM derived from the Rock-Eval analysis could provide the source and early diagenetic information of OM in the reservoir cores. The S1, S2, S3, TOC, and HI show significant decreasing trends with increasing profile depths in the ZT and LA cores, suggesting that the sedimentary OM has either been affected by autochthonous inputs or by extensive degradation (Duan et al., 2015). For the XFJ core, the TOC as well as the other pyrolytic parameters (except HI proxy) show increasing trends with depth, suggesting the degradation and oxidation of OM and/or terrestrial inputs of the OM are the primary factors affecting the variation of OM.

Carbon isotope analyses offer an important tool for identifying the sources of OM in lacustrine sediments. Different primary producers have distinctive carbon isotope compositions. The average $\delta^{13}C$ values of C3 plants is around −27‰ to −26‰, whereas the C4 plants have average $\delta^{13}C$ values of −14‰ to −13‰. Although the $\delta^{13}C$ values in phytoplankton are in a broad range of −17‰ to −45‰ (Boschker et al., 1995), it can be identified by the combination of other proxies (e.g., elemental C/N ratios). All the corrected $\delta^{13}C$ values of sedimentary OM in the three reservoirs vary from −25.5‰ to −19.9‰ (Fig. 1), which are in the range of phytoplankton and C3 plants. However, their corresponding C/N ratios are relatively low than those for higher plants (> 12) (Fig. 1), suggesting the predominant contribution of phytoplankton in the OM of reservoirs. The $\delta^{13}C$ values in ZT sediments are more enriched (average: −20.8‰) than those in LA (average: −24.6‰) or XFJ (average: −24.1‰) sediments, which may be attributed to high phytoplankton productivity (chlorophyll a = 90.7 μg/L), anaerobic sediments with high rates of methanogenesis, and lack of terrestrial carbon inputs in shallow water bodies (Gu and Schelske, 2004). High phytoplankton can enhance isotopic fractionation and result in enrichment of $^{13}C$ in dissolved inorganic carbon. The removal of $^{12}CH_4$ by intensive methanogenesis also leads to the accumulation of $^{13}C$-depleted OM in sediments (Gu and Schelske, 2004).

After correcting for the Suess effect, the OM in sediments becomes more enriched in $^{13}C$ from the bottom to the top of the ZT core (Fig. 1), reflecting a progressive increase in historical productivity, which is consistent with the vertical variations of TOC, C/N, and HI. Similar observations were also found in the LA core from a depth of 16 cm to the surface layer. Therefore, both ZT and LA reservoirs undergo significant increases in primary productivity during the recent years. As shown in Fig. 1, the correction for the Suess effect can result in opposite conclusions regarding the aquatic productivity, based on uncorrected $\delta^{13}C$ values for ZT and LA. Similar results are also observed in Lake Brunnsviken (Routh et al., 2004), Lake Eric (Schelske and Hodell 1995), and deep Lake Tahoe (Chandra et al., 2005), suggesting the importance, in terms of productivity, of the correction for the Suess effect in the recent $\delta^{13}C$ values for lacustrine sediment cores.

For the XFJ reservoir, the corrected $\delta^{13}C$ values increase from depths of 22 to 10 cm and then decrease abruptly to the surface layers, which may result from the biodegradation of OM in aerobic sediments or from a great number of recent terrestrial loading. However, values of C/N ratios in the upper layers of the XFJ core are very low (C/N ≈ 3) and indicate a



contribution of algal origin in the sediments (Fig. 1). Therefore, the decrease of corrected $\delta^{13}C$ at XFJ is mainly due to the
biodegradation of OM under aerobic conditions. The dissolved oxygen content is 8.54 mg/L at a depth of 36 m, which can
provide sufficient oxygen for microbial metabolism (Fig. S2 in the supporting data). The preferential degradation of more
$^{13}C$-enriched organic compounds (e.g., carbohydrates and proteins) would lead to a decrease in the $\delta^{13}C$ values in the residue
OM (Lehmann et al., 2002).

### 4.2 Monosaccharide compositions in sediment cores

As shown in Table. S2, the TCHO concentrations in the ZT and LA cores show significant decrease in the downcore
sediments, which are similar to the vertical profiles of S2, TOC, C/N, corrected $\delta^{13}C$, and HI (Table S1 and S2 in the
supporting data). For the XFJ core, the TCHO and HI still decline in the downcore sediments as do those in the ZT and LA
cores (Table S1 and S2 in the supporting data). In general, the content of TCHO (0.51–6.4 mg/g) in the three reservoirs is
similar to that of a sediment core in the eutrophic French Aydat lake (1.19–4.58 mg/g) (Ogier et al., 2001), which was also
enriched with neutral sugars at the surface layers of the sediment core.
The compositions of neutral sugars in ZT and LA cores show that glucose is the most abundant sugar in these two
reservoir sediments while galactose, mannose, and arabinose are relatively more abundant than rhamnose, xylose, and fucose
(Fig. 2), which is similar to the monosaccharide composition in phytoplankton (Hamilton and Hedges, 1988). Moreover, the
relative abundances of monosaccharides do not vary much in the ZT and LA cores except for the apparent changes of
glucose and xylose at depths of 10, 16, 32, and 34 cm in the LA core. For the XFJ reservoir, the monosaccharide
composition in the upper layers (0–16 cm) of the sediment core also indicates a dominant origin of algal carbohydrates.
However, xylose significantly increases at a few depths (XFJ core: at 18, 24, and 32 cm depths) (Fig. 2). Moreover, glucose
shows an increasing trend correlated with the abundant xylose, especially between the depths of 18 cm and 34 cm. The
pattern of carbohydrate composition in these samples (18–34 cm) is not in agreement with the post-depositional process of
diagenesis as observed in previous studies (Hamilton and Hedges, 1988; Hedges et al., 1994). They found that the diagenesis
process often led to a decrease in glucose along with a corresponding increase in bacteria-derived deoxy sugars (rhamnose
and fucose) in sediment cores. Therefore, these outliers (18–34 cm) might be related to increasing vascular plant input or
hydrological variation at XFJ, as discussed in the following paragraph. .

### 4.3 Source of neutral carbohydrates

Molecular-level diagnostic parameters have often been used to differentiate microbial, planktonic, and terrestrial
sources (Cowie and Hedges, 1984; Guggenberger et al., 1994). Diagnostic carbohydrate parameters and their results are
presented in Fig. S2 and S3 in the supporting data.
The ratios of mannose to xylose could indicate the OM sources derived from phytoplankton, bacteria, gymnosperm, and
angiosperm tissues (Cowie and Hedges, 1984). As shown in Fig. 3 in the supporting data, the values of the Mannose/Xylose
ratios in most of the sediments at ZT and LA range from 1.51 to 2.70 and 1.49 to 3.50, respectively, except at a depth of 8
cm (1.46) in the ZT core and at the depths of 4 cm (1.18), 8 cm (1.05), 10 cm (0.67), 16 cm (8.60), and 28–32 cm (0.6–1.33)
in the LA core. Thus, most of the samples can be identified as a phytoplankton source (1.5–3.5), suggesting the important
contribution of AOM in these two areas. However, most of the Mannose/Xylose ratios are in the range of 0.23–0.87 for
gymnosperm tissues (< 1) at depths deeper than 16 cm in the XFJ sediments, which indicates the presence of terrestrial OM
derived from angiosperm leaves and grasses (Fig. S7)
The above conclusion is also confirmed by the %Xylose$_b$ parameters ('b' represents a value on glucose-free base)
plotted in Fig. 2 and Fig. S3 in the supporting data. %Xylose$_b$ is a useful biomarker to differentiate the type of terrestrial
input (Cowie and Hedges, 1984). In most of the samples at ZT and LA, %Xylose$_b$ is in the ranges of 9.34–11.8 and
7.10–15.8, respectively, except at the depths of 10 cm (23.5), 16 cm (3.02), and 32 cm (26.4) in the LA core. The





low %Xylose$_b$ values (6.2–17.0) indicate the primary phytoplankton origin of neutral sugars at ZT and LA. The high values
at the depths of 10 cm and 32 cm in the LA core might indicate important terrestrial input. Further evidence is also obtained
from %(Arabinose + Galactose)$_b$ plotted in Fig. S3 in the supporting data and from %(Fucose + Rhamnose)$_b$ plotted in Fig.
S2 in the supporting data. The results from the %Xylose$_b$ versus %(Fucose + Rhamnose)$_b$ plots suggest a phytoplankton
origin in reservoir sediments, which is consistent with that reported in the literature (Boschker et al., 1995). As for
the %(Arabinose + Galactose)$_b$ ratios, their values are mostly in the range of 30.7–44.4 in the sediment cores at ZT, LA, and
XFJ, indicating that the sedimentary OM samples are largely derived from phytoplankton with the ratios of 22–47. Only a
few high values at a depth of 16 cm (48) in the LA core and at depths of 8 cm (50.6) and 34 cm (48.9) in the XFJ core are
likely to indicate an additional origin from non-woody angiosperm tissues and grasses, as demonstrated by Cowie and
Hedges (1994). The above result also implies a different origin for neutral sugars in the upper layers of the XFJ core (0–16
cm) than in the lower layers (> 18 cm), which have been increasingly affected by terrestrial input.

In order to support the above conclusions of the sources of OM by neutral sugars in the reservoirs, monosaccharide

concentrations and heavy metal data are compared to each other (Table S3 and S4 in the supporting data). It is found that
almost all monosaccharides (except Xylose at LA and XFJ) are significantly related to heavy metals (e.g., Zn and Cu) in the
sediment cores of ZT and LA, and the upper layers of the XFJ core (0–16 cm) (Table S4 in the supporting data). However,
only galactose, mannose, and fucose are positively correlated with Zn and Cu in the core of XFJ (0–34 cm), although the
lower layers are increasingly affected by allochthonous input, as discussed in the above paragraphs. As Zn and Cu are
essential nutrients for phytoplankton growth, these relationships provide more evidence for the important contribution of
AOM to carbohydrates in the investigated sediments.

### 273    4.4    Diagenesis of neutral carbohydrates

Carbohydrates are not only useful in identifying the sources of OM but also in evaluating early diagenetic processes

occurring in the post-depositional environment. The four parameters, deoxysugars/pentoses (deoxy S/C5) ratio, glucose
content (mol% or wt%), %(Fucose + Rhamnose)$_b$, and carbohydrate yield (%) in the sediment cores are often used to
evaluate diagenetic changes of OM (Cowie and Hedges, 1984; Ittekkot and Arain, 1986; Opsahl and Benner, 1999; Benner
and Opsahl, 2001; Kaiser and Benner, 2009).

Glucose content is an important factor used to assess the degradation state of OM. Glucose accounts for 58 to 90% of

the carbohydrates in fresh plankton and terrestrial tissues (Cowie and Hedges, 1984; Opsahl and Benner, 1999; da Cunha et
al., 2002). Hernes (1996) proposed that relative mol% glucose in particulate OM could be used as a diagenetic indicator for
organic material in the equatorial Pacific region. In this investigation, wt% glucose in the sediments ranges from 22 to 29.1%
at ZT, from 23.3 to 39.2% at LA, and from 19.6 to 50.3% at XFJ (Fig. 3), suggesting that neutral sugars are biodegraded in
the sedimentary OM. This conclusion is also confirmed by the carbohydrate yield (%), which usually represent 30–40% of
TOC in fresh tissues of plant and phytoplankton but less than 9% in sediments (Cowie and Hedges, 1984; Opsahl and
Benner, 1999). Carbohydrate yields range from 1.93 to 13.53% in the sediments of the three reservoirs (Fig. 3). It is also
suggested that neutral sugars degrade significantly in the investigated sediments. These results are consistent with the
general observation from previous studies (Ogier et al., 2001). In general, the carbohydrates in the reservoir sediments are
extensively transformed and degraded. However, the stability of their compositions was found in the downcore sediments.
Whether the carbohydrate compounds are degraded mainly in the water column or in the sediment core will be discussed
below (Fig. 3).

Keil (1998) found that the %wt(Fucose + Rhamnose)$_b$ values could reflect the diagenesis process of neutral

carbohydrates. This index was elevated as the sediment particle sizes decreased, suggesting that smaller size fractions
showed a higher degree of degradation. Their observation was also consistent with other diagenetic indices such as lignin
and non-protein biomarkers (Keil et al., 1998). In this study, the values of %wt(Fucose + Rhamnose)$_b$ at three reservoirs do





not vary significantly with the sediment core, and there is no decline in wt% glucose with a corresponding increase of %wt
(Fucose + Rhamnose)$_b$ in each of the ZT, LA, and XFJ sediment cores (Fig. 3), suggesting that the process of degradation
occurs mainly during the settling period rather than after deposition. Further evidence in support of this conclusion can be
obtained from the ratio of deoxy sugars (e.g., rhamnose and fucose) to C5 (e.g., arabinose and xylose) (deoxy S/C5). The
deoxy S/C5 ratios also remain almost unchanged throughout the sediment cores of ZT, LA, and XFJ (Fig. 3). Therefore,
although sinking organic matter-rich particles and their carbohydrates in these reservoirs suffer from intensive oxidation and
degradation in the water column during their transit to bottom sediments, some fractions are selectively preserved in the
sediment cores and remain almost unchanged during post-deposition, as observed before (Cowie and Hedges, 1984; Moers et
al., 1990; Hicks et al., 1994).
Carbohydrates are derived not only from storage polymer but also from the cell membrane of phytoplankton. The
monosaccharide residues in both kinds of polymers are bound to each other via glycosidic bonds (Cowie et al., 1992; Cowie
and Hedges, 1994; Hernes et al., 1996). In general, glucose is bound mainly in an unbranched, starchlike β-1,3-glucan
storage polymer (Handa et al, 1969), whereas mannose, galactose, xylose, fucose, and rhamnose are characteristically more
abundant in the cell walls (Cowie et al, 1992). The cell-wall aldoses are bound primarily in branched and often
heterogeneous structural polymers. On the contrary, glucose, arabinose, and ribose are relatively concentrated in intracellular
polysaccharides. In the present study, concentrations of mannose, galactose, arabinose, and xylose on a glucose-free base are
dominant in the sediment cores, and the carbohydrate compositions are mostly unchanged. Hence, the neutral carbohydrates
in the sediment cores are largely related to structural polysaccharides, which is more resistant to microbial degradation than
storage polysaccharides.
In support of the above conclusion, the $k$ values of deoxy S/C5 were calculated using a "multi-G" model (Wang et al.,
1998) to evaluate neutral sugar degradation. The k value is 0.0025 yr$^{-1}$ for ZT, 0.0021 yr$^{-1}$ for LA, and 0.0025 yr$^{-1}$ for XFJ. It
is found that the decomposition of 95% neutral sugar in sediments will take thousands of years, which is similar to the results
of TCHO in the ocean sediments (Wang et al., 1998) and the degradation of OM in the lacustrine sediments (Li et al., 2013).
The increasing downcore trends of glucose and OM in the XFJ core are different from those in the ZT and LA cores,
suggesting that the variation of OM and neutral sugars are site-independent and may be related to the different trophic states,
the various sources of OM, and the hydrological changes in different depositional environments. Moreover, the downcore
OM profiles in some of the sediment cores have not exhibited decreasing trends (Kirk et al., 2011, Meyer, 1997), which are
not consistent with the traditional degradation model and mechanism. Hence, more work would be needed for investigating
the sources, compositions, and early diagenesis of carbohydrates in different aquatic environments.
### 4.5   Effects of climate change on primary productivity and carbohydrates
Carbohydrates are important organic components from aquatic algae and have undergone extensive degradation during
settling. However, large amounts of resistant structural carbohydrates (e.g., cell walls) containing source information can be
preserved in the sinking particles and sediments. Moreover, HI has been widely utilized as a useful indicator of primary
productivity during recent years (Gasse et al., 2001; Stein et al., 2006; Bechtel and Schubert, 2009). As shown in Fig. 4 and
Table S4 in the supporting data, HI values in the ZT and LA cores are positively correlated with monosaccharides, especially
the algae-dominated, galactose, mannose, fucose, and arabinose (Ittekkot and Arain, 1986; Hamilton and Hedges, 1988;
D'souza et al., 2003), which are usually dominated in structral cell walls of planktonic algaes (Hecky et al., 1973; Haug et al.,
1976). For the XFJ core, significant correlations are also found between the monosaccharides and the HI values, except for
rhamnose, fucose, and xylose. It is noted that the a few samples at depths of 8–10 cm and 30–34 cm in the LA core and at
depths of 18–34 cm in the XFJ core are excluded due to the inputs of allochthonous OM to the sediment cores. Therefore,
monosaccharides (e.g., galactose and mannose) can be used for the reconstruction of historical productivity in the subtropical
reservoirs.





As shown in Table S4 in the supporting information, the contents of glucose, galactose, arabinose, mannose, fucose,
and rhamnose are also positively correlated with S2 and HI values in the ZT, LA, and XFJ sediment cores. However, each of
them shows no/or weak correlations with the diagenetic parameters of neutral sugars and OM (e.g., % (Fucose + Rhamnose)[b],
Deoxy S/C5). Moreover, the xylose concentrations representing terrestrial inputs do not show significant correlations with
the productivity proxies (e.g., S2 and HI) or any of the other monosaccharide concentrations at LA and XFJ. Thus, the above
evidence supports that the increasing neutral sugars in the reservoir sediments are mainly attributed to algal productivity
rather than the degradation of neutral sugars during post-diagenesis.
In order to understand the effects of climate change on the historical variations of primary productivity, the $T_5$ values
over 60 years are compared with the profiles of carbohydrates in the three reservoirs (Fig. 5 and Table S4). The
monosaccharide profiles at ZT and LA show good correlation with $T_5$ during the past 60 years (for ZT, Galactose:$T_5$, $R^2$ =
0.824, $p < 0.01$; Mannose:$T_5$, $R^2 = 0.824$, $p < 0.01$; Fucose:$T_5$, $R^2 = 0.805$, $p < 0.01$; for LA, Galactose:$T_5$, $R^2 = 0.885$, $p <$
0.01; Mannose:$T_5$, $R^2 = 0.699$, $p < 0.01$; Fucose:$T_5$, $R^2 = 0.883$, $p < 0.01$;), suggesting that an increase in temperature likely
enhances the deposition of carbohydrates in sediments (Fig. 5 and Table S4). Moreover, total nitrogen (TN) and total
phosphorus (TP) showed weaker correlations with $T_5$ than carbohydrates for the sediment cores of ZT and LA (Table S5;
Duan et al., 2015). The TN and TP concentrations can be used to reflect the historical inputs of nutrients in the ZT and LA
reservoirs. Furthermore, the TP and TN concentrations at ZT and LA remained at a low level (mostly TP < 0.1 mg/g, TN <
0.4%) during the past six decades and are far lower than those in sediments of other eutrophic reservoirs (Duan et al., 2015).
Further evidence can be found in the results from principal component analyses (PCA) in the Fig. S8 in the supporting data.
The $T_5$, HI, and monosaccharides are in the first principal component and account for 76.5% and 67.3% of the total variance
in the LA and XFJ reservoirs, whereas the second principal component of TP and TN accounts for 8.36% and 11.7% of the
total variance, respectively. Hence, the nutrient input is not the key factor affecting carbohydrates in the three reservoirs. In
addition, the factor of light can also be excluded by the records of the annual hours of daylight, which show progressively
decreasing trends from 1950 to 2010 at Guangzhou and Heyuan (Fig. S5). In conclusion, the increase in deposition of
carbohydrates in sediments at ZT and LA corroborates very well with the increase in temperature (Fig. 5). For the XFJ
reservoir, the profiles of monosaccharides also show a positive correlation with the temperature variations (for XFJ,
Galactose:$T_5$, $R^2 = 0.702$, $p < 0.01$; Fucose:$T_5$, $R^2 = 0.744$ $p < 0.01$;), but show no relationship with TN or TP concentrations
(Fig. 5, Table S4). Although the primary productivity is lower in the XFJ reservoir than in the ZT or LA reservoirs, it is still
significantly affected by the increasing temperature. Although the TOC and S2 values have declined with increasing
temperature in the XFJ core profile due to the phosphorus-limited trophic level, the neutral sugar contents still increase (Fig.
5). Therefore, the above results suggest that neutral sugar in the ZT, LA, and XFJ reservoirs is strongly associated with
climate warming in the subtropical area.
**5   Conclusions**
The sources, composition, and diagenesis of carbohydrates in the sediment cores of three subtropical reservoirs were
investigated by conducting acid hydrolysis coupling with high-performance liquid chromatography (HPLC) with pulsed
amperometric detection (PAD). Glucose, mannose, and galactose are the most abundant monosaccharides. Monosaccharide
composition and diagnostic parameters (mannose/xylose ratio, arabinose plus galactose, xylose) indicate a predominant
contribution of phytoplankton in the whole sediment cores of the ZT and LA reservoirs and in the upper layers of the XFJ
core (0–16 cm). The carbohydrates are partially degraded during the settling. It is found that the degradation proxies
(S2/RC, %wt (fucose + rhamnose)$_b$, and deoxy S/C5) have not varied much in all of the whole sediment cores.
The corrected $\delta^{13}$C and C/N ratios in sedimentary OM can be used to reflect the historical changes of productivity in the
subtropical reservoirs. Based on the higher corrected $\delta^{13}$C values and the lower C/N ratios, increased productivity was





observed in the upper layers of ZT and LA reservoirs, which are consistent with the increase of monosaccharides (e.g.,
galactose, mannose, fucose, and arabinose) and HI. As for XFJ reservoir, the corrected $\delta^{13}$C values abruptly decrease with very
low C/N ratios, which may indicate different sources and biodegradation in the underlying sediments. Moreover, strong
positive correlation between TCHO and HI is found both in the mesotrophic reservoirs (ZT and LA) and in the oligotrophic
reservoir (XFJ) in this investigation, suggesting that TCHO is related to primary productivity in the studied subtropical
reservoirs. Furthermore, increasing levels of carbohydrates in the three reservoir cores show significant relationships with $T_5$
during the last 60 years. Elevated temperatures lead to increasing levels of carbohydrates in the sediment profiles during last
six decades. Therefore, this investigation provides important evidence for the effect of climate change on the aquatic
ecosystems in the low latitude region. To further develop the productivity indicator of carbohydrates, more work is needed to
improve the detection of all the sedimentary monosaccharides (e.g., ribose) during acid pyrolysis.
**Author contribution**
Jing'an Chen and Yong Ran designed the experiments and Dandan. Duan, Dainan Zhang, Jingfu Wang, and Yu Yang carried
them out. Dandan Duan prepared the manuscript with contributions from all co-authors.
**Competing interests**
The authors declare that they have no conflict of interest.
**Acknowledgments**

Data supporting Figures S1–S5 are available as Tables S1, S2, S3, and S4 in Supporting Information. The annual

average air temperature and five-year moving averages of the air temperature on the time scale of 60 years in the Guangzhou
area and Heyuan area were obtained from the China Meteorological Data Sharing Service System (CMDSSS).

This study was supported by a key project of NNSFC-Guangdong (U1201235), a project of the National Natural

Science Foundation of China (41473103), and a project of the Earmarked Foundation of the State Key Laboratory
(SKLOG2015A01). We thank Professor Ronald Benner and Dr. Michael Philben at the University of South Carolina for help
with the analysis of neutral sugars conducted in Benner's laboratory. This is contribution No. IS-2000 from GIGCAS.



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





**Figure captions**
**Figure 1**. Vertical variations of original $\delta^{13}$C, corrected $\delta^{13}$C, and C/N in sediment cores of three reservoirs
**Figure 2.** Profiles of neutral carbohydrates in sediment cores of the reservoirs
**Figure 3.** Vertical profiles of yield (%), Deoxy S/C5, wt% Glucose and wt % (Fucose+Rhamnose)$_b$ in the ZT, LA, and
XFJ reservoirs
**Figure 4.** Relationship of HI with single sugar compounds in the ZT, LA, and XFJ reservoirs (For LA, the sample at 8-10 cm
depths and at 30-34 cm depths were excluded; For XFJ, samples below 16 cm were excluded). The HI data are coted from
Duan et al. (2015).
**Figure 5.** Temporal profiles of five-year moving temperature ($T_5$), hydrogen index(HI) and single neutral sugars in sediment
cores from the three reservoirs.







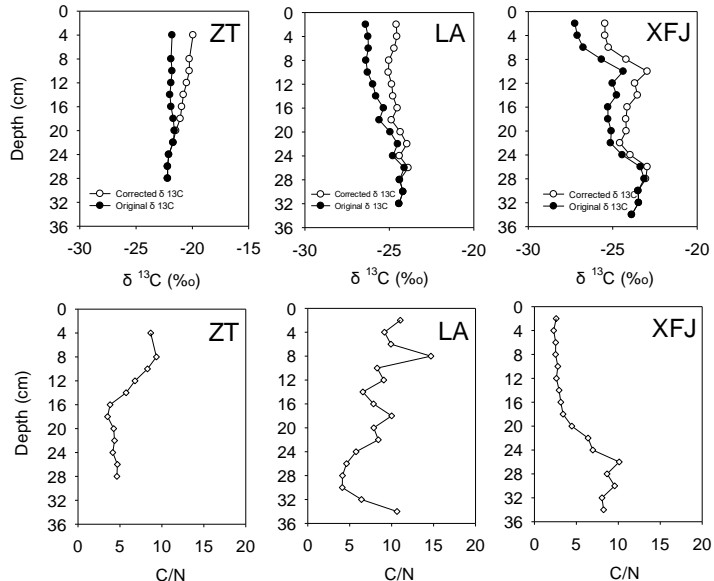


**Figure 1**. Vertical variations of original $\delta^{13}$C, corrected $\delta^{13}$C, and C/N in sediment cores of three reservoirs










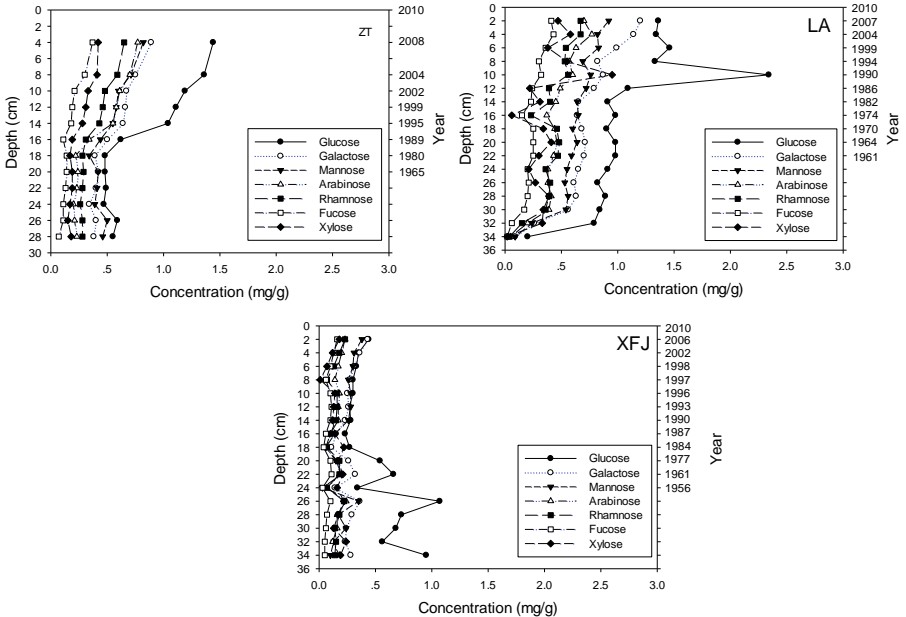


**Figure 2.** Profiles of neutral carbohydrates in the sediment cores of the reservoirs








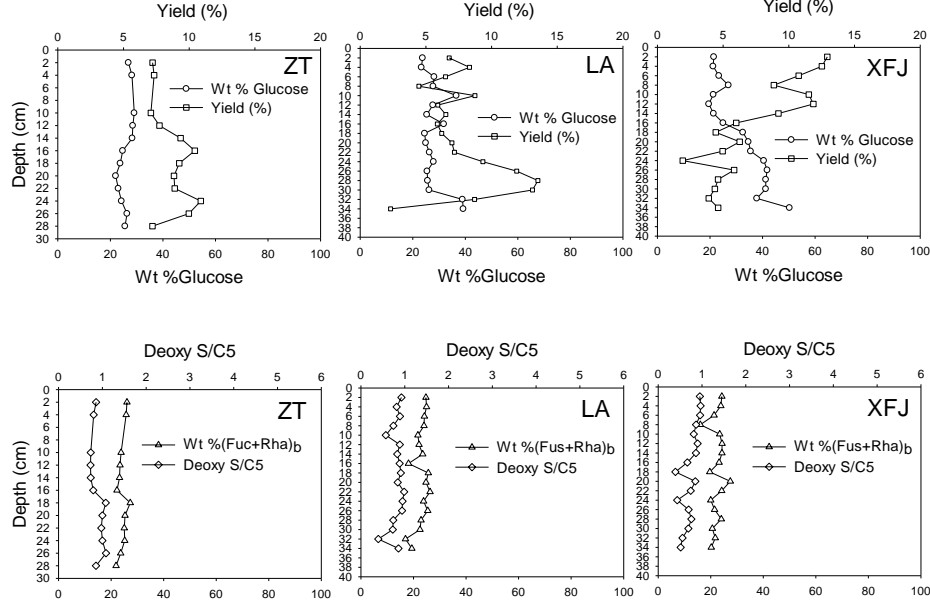




**Figure 3.** Vertical profiles of yield (%), wt% glucose, deoxy S/C5, and wt% (Fucose+Rhamnose)ᵦ at the ZT, LA, and XFJ

reservoirs
























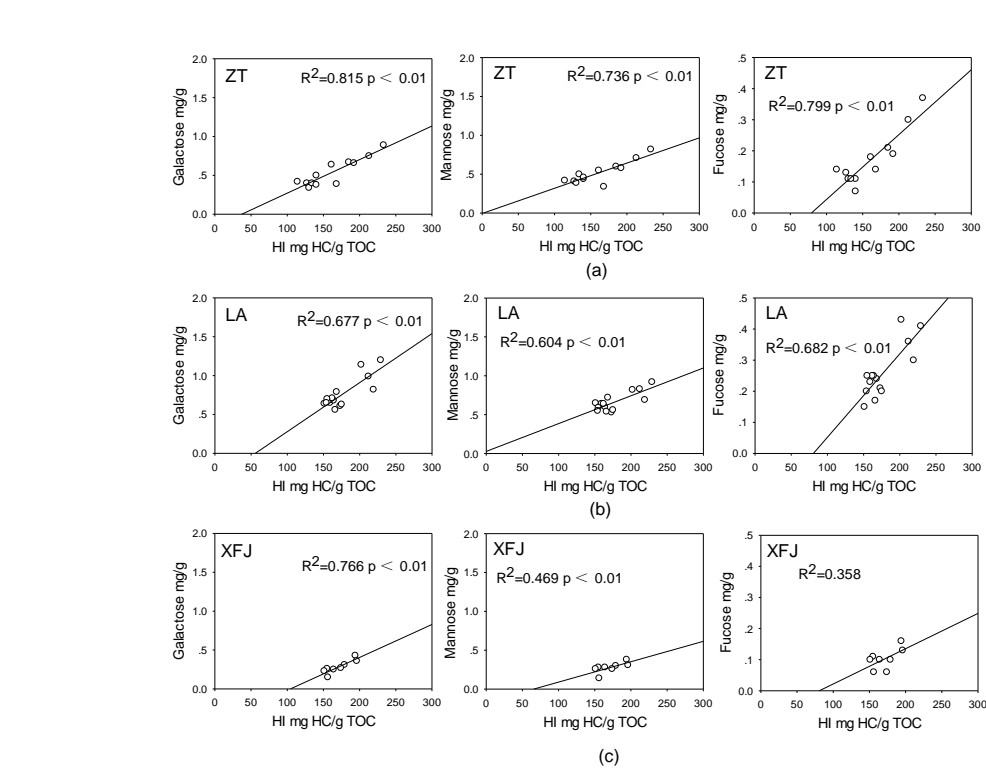

**Figure 4**. Relationship of HI with galactose, mannose, and fucose in the ZT, LA, and XFJ reservoirs (For LA, the sample
at 8-10 cm depths and at 30-34 cm depths were excluded; For XFJ, samples below 16 cm were excluded). The HI data
were cited from Duan et al. (2015).






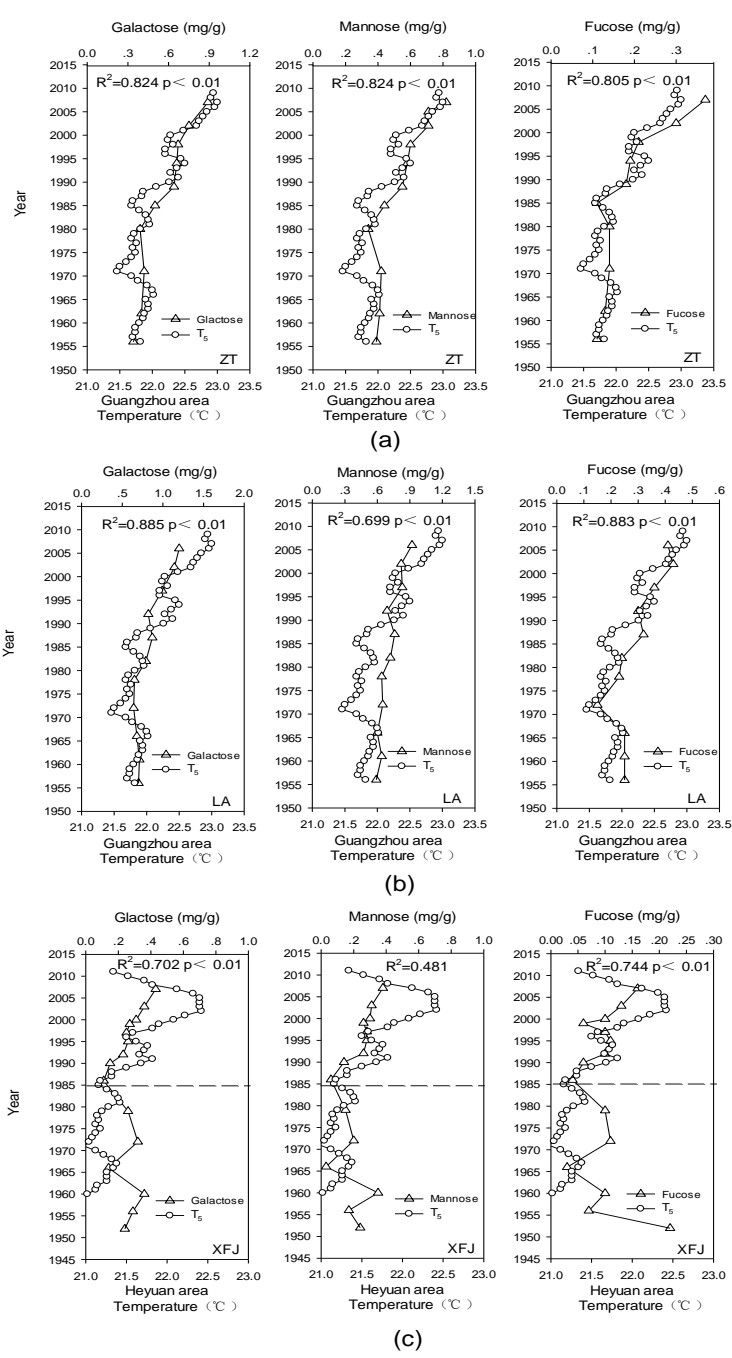

**Figure 5**. Temporal profiles of five-year moving temperature ($T_5$) and single neutral sugars in sediment cores from the three reservoirs.