# Peer review of "of subtropical reservoirs, South China"

_Biogeosciences, 2016_

## Referee Comment (RC1) · Anonymous Referee #1 · 4 Jan 2017

Reviewers' comments:

Duan et al., have collected sediment cores from three tropical reservoirs in South China and analyzed them for the neutral carbohydrates along with algal organic matter (AOM) content, carbon isotopic composition, and elemental C/N ratios. Based on these data, they investigate the source, composition and diagenesis of the neutral carbohydrates and their relationships with the history of algal productivity induced by climate change over the last 60 years. This manuscript presents interesting results and requires minor revision before it can be accepted for publication. The following comments might help authors in their revising:

1. First, the manuscript needs some cohesive discussion to emphasize more on the

combination uses of carbon isotopic composition, pyrolytic organic parameters and carbohydrates composition.

2. Second, the correlation between air-temperature changes in South China over 60 years and the trend in organic parameters of the three studied lakes is interesting. This part might be worth exploring in the future. Thus, a research outlook could be given by the authors.

Specific comments/questions

Line 24: The "single neutral carbohydrate" should be replaced by "monosaccharide", please correct it all throughout the text.

Line 47: The lowercase delta notation for isotopes should be italics. Please correct these all throughout the text.

Line 94: The isotopic values are reported relative to the V-PDB Belemnite Standard, not just PDB.

Line 108: Is the "AC 5OW-X8" right? According to the Michael's paper in 2015, the cation resin should be "AG 50W-X8".

Line 243: "Fig. 3" should be "Fig. S3".

Line 355: "Fig. S8" should be "Fig. S4".

---

## Referee Comment (RC2) · Anonymous Referee #2 · 3 Mar 2017

**Source, composition, and environmental implication of neutral carbohydrates in sediment cores of subtropical reservoirs, South China**

This paper by Duan et al. compared organic matter characteristics (d13C, C/N) and monosaccharide distributions in sediment cores from three lakes with different depths (3 m, 17 m, and 36 m) and trophic states (mesotrophic vs. oligotrophic). The neutral sugar data is nicely presented and discussed in the context of source and changes in productivity and climate, making this a useful addition to the field. However, connections between the carbon isotopic data and the neutral sugars are not clear in the text though correlation between them is mentioned in the abstract and

displayed in table S4. The manuscript would benefit from expanding on the utility of combining these types of measurements rather than discussing the data and their implications separately. After revising this and the minor (but numerous) issues below, I would recommend the paper for publication in Biogeosciences. In addition to these comments, the manuscript should be checked carefully for small grammatical errors such as missing or incorrect articles and singular/plural subject/verb issues.

**Minor comments:**

- pg 2, lines 44-48: Phytoplankton is plural so the verbs should be 'remove,' 'deplete,' and 'discriminate.' Line 48 should be values.
- pg 2, line 52: O'Reilly et al. (2005)
- pg 2, line 53: Verburg reference should be 2007
- pg 2, line 55: It is not clear what 'it' in this sentence is referring to, please revise
- pg 3, line 58: Kirk et al. (2011)
- pg 3, lines 66-67: Typo, add 'in' after 'help'; also 'Besides' is not correctly used here, please revise
- pg 4, line 94: is this actually V-PDB?
- pg 4, line 94: from where is 'Product ID: GBW 04408' sourced?
- pg 4, lines 107-113: Michael et al., 2015 is not listed in the references
- pg 6, line 192: Also not clear what 'it' refers to in this sentence, please clarify
- pg 7, line 243: In this section (and in a few other places throughout the manuscript) the monosaccharide names are strangely capitalized?
- pg 8, line 281: Hernes et al. (1996)
- pg 8, line 292: Keil et al. (1998)
- pg 9, line 308: Should be no 'et al.,' for Handa, 1969 reference
- pg 9, line 316: Another unclear 'it' usage, please revise
- pg 9, line 329: Gasse et al. should be 1991 as listed in the references
- pg 9, lines 331-332: The use of 'algae-dominated' and then 'usually dominated

in . . . algaes' is redundant. Additionally, the wording of 'dominated in' as a verb is grammatically incorrect (perhaps 'are usually dominant in'?) and 'algaes' is plural without the 's'

- pg 9, line 332: should be Haug and Myklestad, 1976
- pg 9, line 334: typo '. . . the a. . .'; remove either 'the' or 'a'
- pg 10, line 340: the / between 'no/or' is not needed; alternatively the 'or' could be removed ('no/weak correlations')
- pg 10, line 367: this should be changed to 'neutral sugars . . . are'
- pg 11, line 379: change 'are' to 'is'
- pg 11, line 385: insert 'the' before 'last six decades'
- Figure 1: Is it possible to use the same scale for all three isotope profiles? Perhaps with a range from -28 to -18 so that the reader can easily compare the three sites visually
- Figure 2: The concentration range on the x-axis is quite large for the data, making it difficult to see variations with depth. Aside from the single outlier in the LA glucose profile, could these be changed to more appropriate ranges for the data?
- Borch et al. 1997, Gu et al. 2004, Kaiser and Benner 2000, Marchand et al. 2008, Philben et al. 2015, Mopper et al. 1992, Ran et al. 2007, and Wakeham et al. 1997 are listed in the references but not cited in the text.

---

## Referee Comment (RC3) · Anonymous Referee #3 · 26 Apr 2017

Duan et al. obtained a very nice, enriched, dataset including neutral sugars and other parameters in three subtropical reservoirs. Based on the concentrations and composition of the neutral sugars, isotope values of TOC, and C/N ratios, they investigated source and diagenesis pathways of sedimentary organic matter (SOM). They concluded that the dominant source of SOM was phytoplankton in the ZT, LA and upper XFJ reservoirs, and there was not much degradation of carbohydrates downward in the sediment cores. Also, there seems to be a nice correlation between temperature and the levels of carbohydrates over the past 60 years. I think this paper would be of interest to the community and worthy of being published, but I have issues with the way they presented, too broad and without a clear focus. The authors discussed a lot of

possible sources and phytoplankton among different reservoirs, but they did not even mention why different patterns, ZT and LA vs. XFJ, were observed,. In addition, some of the conclusions are very speculative. Overall, I do not feel this paper is ready without a major revision.

The section of Materials and Methods needs more work. They need to include the information about measuring the sedimentation rate and pyrolysis. I know they have these in the Duan et al. 2015 paper, but these should be briefly described, since they use those data in the Results section and you can't force the audience to read your other paper. It is unclear how many cores they collected. In other words, how representative are these cores to the whole reservoirs. If these systems have been impacted by human activities, such as dredging, sediments in these reservoirs can be very heterogeneous.

A main issue with the manuscript is the lack of focus on the discussion. They talked about a lot of difference topics, but it was written like a result section with titles like, "OM characteristics", "Monosaccharide composition", "Source of neutral carbohydrates", and so on. In other words, it reads more like a data report rather than a research paper.

Line 43: "offer"

Line 49: delete "and impacted"

Line 54: any evidence about the Suess effect would be particularly stronger in the industrialized areas such as Pearl River Delta? I would assume this should be about the same worldwide considering the fast CO2 mixing in the air.

Line 127: awkward wording, should be "productivity significantly contributed to dissolved oxygen content"

Line 130: nutrients levels are always higher in the deeper depth. What do you mean by "be brought" to deeper depths"?

Line 136: again, describe the pyrolysis

Lines 193-196: have to be careful about the C/N ratios. Decomposition of terrestrial organic matter can decrease C/N ratios, not necessarily source related. This has been well documented in composting studies. Also, the C/N ratios of 3 in the XFJ upper layers should be interpreted in a more careful way. I don't think you can simply say "algal origin", because C/N ratios fresh algae are typically about 6-7, and even pure bacterial biomass typically have C/N ratios of 4. It is not very clear how you would get SOM with such low C/N ratios.

Line 200: the removal of CH4 (13C light) should lead to the accumulation of 13C-heavey SOM

Line 214-216: too speculative. The DO level you mentioned refers to the water, not sediment. I think the major OM decomposition in these OM-enriched sediments is through anaerobic pathway, unless you have DO profile data in sediment cores.

Line 270: it's interesting to note the correlations between Zn and Cu and carbohydrates. I think more data analysis is needed, such as the contents of Zn and Cu in algae and how they trace metal got preserved, etc. It's not enough to simply have a correlation and then argue they were from phytoplankton. For example, it could have been sourced from industry contamination.

Section 4.4. When the individual carbohydrates are normalized to TOC, I don't think there is much a decreasing trend at all (Table S2). In other words, carbohydrates simply are not good indicators of digenesis. This section should be strongly condensed.

Section 4.5. This section is interesting, but still at a speculative stage. Issues why we would expect carbohydrate increase, such as increased phytoplankton production or decomposition of SOM under warmer climate?

---

## Author Comment (AC2) · 24 Jun 2017

Title: Source, composition, and environmental implication of neutral carbohydrates in sediment cores of subtropical reservoirs, South China Authors: Dandan Duan, Dainan Zhang, Yu Yang, Jingfu Wang, Jian'an Chen, and Yong Ran\* doi:10.5194/bg-2016-505

Dear Editor and reviewer:

Thank you very much for your comments on our manuscript. We have carefully revised the manuscript according to your valuable and helpful comments. Our responses are marked in blue color below.

We are looking forward to your further comments and decision.

With best regards!

Dr. Yong Ran

**—— Anonymous Referee #3:**

Duan et al. obtained a very nice, enriched, dataset including neutral sugars and other parameters in three subtropical reservoirs. Based on the concentrations and composition of the neutral sugars, isotope values of TOC, and C/N ratios, they investigated source and diagenesis pathways of sedimentary organic matter (SOM). They concluded that the dominant source of SOM was phytoplankton in the ZT, LA and upper XFJ reservoirs, and there was not much degradation of carbohydrates downward in the sediment cores. Also, there seems to be a nice correlation between temperature and the levels of carbohydrates over the past 60 years. I think this paper would be of interest to the community and worthy of being published, (1) but I have issues with the way they presented, too broad and without a clear focus. The authors discussed a lot of possible sources and phytoplankton among different reservoirs, but they did not even mention why different patterns, ZT and LA vs. XFJ, were observed,. In addition, some of the conclusions are very speculative. Overall, I do not feel this paper is ready without a major revision.

Reponses: We have made some revisions on the discussion in order to refine a clear focus: The combination uses of neutral sugars, carbon isotopic composition, and pyrolytic organic parameters are recommended for reflecting the historical changes of productivity in subtropical reservoirs. They can also be used for the investigation of climate change effects on algal productivity in these reservoirs. We also added detailed discussion on the causes and reasons for the possible sources among different reservoirs. It is related to not only native species of algae, plant, and bacteria, but also to historical changes of hydrological conditions, nutrient level, anthropogenic activities,
**and so on.**

(2) The section of Materials and Methods needs more work. They need to include the information about measuring the sedimentation rate and pyrolysis. I know they have these in the Duan et al. 2015 paper, but these should be briefly described, since they use those data in the Results section and you can't force the audience to read your other paper. It is unclear how many cores they collected. In other words, how representative are these cores to the whole reservoirs. If these systems have been impacted by human activities, such as dredging, sediments in these reservoirs can be very heterogeneous.

Reponses: We have added the information of measurements for the sedimentation rate and pyrolysis according to your suggestion.

We have sampled 2 or 3 cores for each reservoir and all of the sediment cores were collected in the center of these reservoirs. Moreover, the reservoirs are mainly suppled by rainfall and are far away from the industrial center. The aquaculture is forbidden and there is no dredging activities in the investigated areas.

(3) A main issue with the manuscript is the lack of focus on the discussion. They talked about a lot of difference topics, but it was written like a result section with titles like, "OM characteristics", "Monosaccharide composition", "Source of neutral carbohydrates", and so on. In other words, it reads more like a data report rather than a research paper.

Reponses: The object of this study is to validate the combined uses of the carbon isotopic composition, pyrolytic organic parameters, and neutral sugars as the potential proxies for historical changes of productivity in subtropical reservoirs and their relationships with the climate changes in the investigated areas. The section of "OM character-istics" was written for the applicability of pyrolytic organic parameters as algal proxies. Both "Monosaccharide composition" and "Source of neutral carbohydrates" sections were compiled for the applicability of neutral sugars as algal proxies in the investigated section.
areas. We have made some revisions to emphasize a clear focus according to your suggestion.

Specific comments:

Line 43: "offer"

Reponses: We have changed "offers" to "offer" in the manuscript according to your suggestion.

Line 49: delete "and impacted"

Reponses: We have deleted the "and impacted" in the manuscript according to your suggestion.

Line 54: any evidence about the Suess effect would be particularly stronger in the industrialized areas such as Pearl River Delta? I would assume this should be about the same worldwide considering the fast CO2 mixing in the air.

Reponses: With rapid economic and industrial development in the Pearl River Delta, the local lakes and reservoirs is more easily affected by the Suess effect due to the high-emssion of COÂň2 even the CO2 mixing is fast in the air. However, there is no data for the Suess effect in this area. Therefore, we have made some revision on the description of the Suess effect in the manuscript according to your suggestion.

Line 127: awkward wording, should be "productivity significantly contributed to dissolved oxygen content"

Reponses: We have changed "the important role of oxygen content in the growth of productivity" to "productivity significantly contributed to dissolved oxygen content" in the manuscript according to your suggestion.

Line 130: nutrients levels are always higher in the deeper depth. What do you mean by "be brought" to deeper depths"?

BGD
Reponses: During winter, the top layers of the lake have relative higher levels of productivity and the bottom layers have higher contents of nutrient. Moreover, the water column mixes from top to bottom in the lake due to the decrease in temperature (so-called autumn overturn). Therefore, the relative high contents of nutrient can be transported by the water flow to the upper depths, resulting in the increase of nutrient and productivity in the entire water column. We have made some revisions in the manuscript according to your suggestion.

Line 136: again, describe the pyrolysis

Reponses: We have added the instruction and description of pyrolysis in the manuscript according to your suggestion.

Line 193-196: have to be careful about the C/N ratios. Decomposition of terrestrial organic matter can decrease C/N ratios, not necessarily source related. This has been well documented in composting studies. Also, the C/N ratios of 3 in the XFJ upper layers should be interpreted in a more careful way. I don't think you can simply say "algal origin", because C/N ratios fresh algae are typically about 6-7, and even pure bacterial biomass typically have C/N ratios of 4. It is not very clear how you would get SOM with such low C/N ratios. The very low C/N ratios are likely to be related inorganic N in minerals. As the TOC contents are quite low in XFJ, their inorganic N contents will affect the C/N ratios. We will account for this effect in the revised manuscript.

Line 200: the removal of CH4 (13C light) should lead to the accumulation of 13C heavy SOM

Reponses: We have changed "The removal of 12CH4 by intensive methanogenesis also leads to the accumulation of 13C-depleted OM" to "the removal of CH4 (13C light) should lead to the accumulation of 13C heavy SOM" in the manuscript according to your suggestion.

Line 214-216: too speculative. The DO level you mentioned refers to the water, not

BGD
sediment. I think the major OM decomposition in these OM-enriched sediments is through anaerobic pathway, unless you have DO profile data in sediment cores.

Reponses: We have deleted the speculative part and rewrite the paragraph in the manuscript according to your suggestion.

Line 270: it's interesting to note the correlations between Zn and Cu and carbohydrates. I think more data analysis is needed, such as the contents of Zn and Cu in algae and how they trace metal got preserved, etc. It's not enough to simply have a correlation and then argue they were from phytoplankton. For example, it could have been sourced from industry contamination.

Reponses: We don't have Zn and Cu data in algae from the investigated areas. However, the Pb contents in the sediments of these reservoirs are very low, which suggesting there is no or little industry contamination in this area.

Section 4.4. When the individual carbohydrates are normalized to TOC, I don't think there is much a decreasing trend at all (Table S2). In other words, carbohydrates simply are not good indicators of digenesis. This section should be strongly condensed.

Reponses: We have condensed the section 4.4 in the manuscript according to your suggestion.

Section 4.5. This section is interesting, but still at a speculative stage. Issues why we would expect carbohydrate increase, such as increased phytoplankton production or decomposition of SOM under warmer climate?

Reponses: We have observed significant correlations among T5 temperaure and contents of algal monosacchrides in the investigated reservoirs. Moreover, these monosacchrides are significantly related to algal parameters (e.g. HI and S2). However, the diagenesis processes of neutral sugars and OM are estimated to be quite slow in the bottom sediments. Some fractions could be selectively preserved and remain unchanged during the post deposition. Therefore, the productivity proxies derived

BGD
from the neutral sugars could be significantly related to the climate warming.

---

## Author Response (AR1)

**The point-by-point reply to the reviewers' and editor's comments:** (Our responses are marked in blue color)

Title: **Source, composition, and environmental implication of neutral carbohydrates in sediment cores of subtropical reservoirs, South China**

Authors: Dandan Duan, Dainan Zhang, Yu Yang, Jingfu Wang, Jian 'an Chen, and Yong Ran*

doi:10.5194/bg-2016-505

Dear Editor:

We'd like to appreciate you and three reviewers very much for the helpful and thoughtful comments on our manuscript. We have carefully revised the manuscript according to your suggestions and the reviewers' comments. The point-by-point responses are written in blue color. For your convenience, the annotated version of the manuscript is loaded with the final version of the manuscript. We sincerely appreciate your consideration.

With best regards,

Dr. Yong Ran
* * *
**Anonymous Referee #1:**

Duan et al., have collected sediment cores from three tropical reservoirs in South China and analyzed them for the neutral carbohydrates along with algal organic matter (AOM) content, carbon isotopic composition, and elemental C/N ratios. Based on these data, they investigate the source, composition and diagenesis of the neutral carbohydrates and their relationships with the history of algal productivity induced by climate change over the last 60 years. This manuscript presents interesting results and requires minor revision before it can be accepted for publication. The following comments might help authors in their revising:

1. First, the manuscript needs some cohesive discussion to emphasize more on the combination uses of carbon isotopic composition, pyrolytic organic parameters and carbohydrates composition.

**Response**:

We have added some cohesive discussion on the interrelationship and combination uses of the carbon isotopic composition, pyrolytic organic parameters and carbohydrates composition.

As shown in the Table S4, positive correlations between corrected $\delta^{13}C$ values, hydrogen index (HI), monosaccharide contents of algal origin, and five- year moving average temperature ($T_5$) were only observed in ZT core. The corrected $\delta^{13}C$ at LA and XFJ cores showed no relationship with other productivity parameter (HI) and $T_5$ even though the corrected $\delta^{13}C$ were enriched in the upper layers of LA core. This may result from the effects of organic matter degradation and variable terrestrial inputs, etc. on the carbon isotopic composition ($\delta^{13}C$) at mesotrophic and oligotrophic reservoirs (LA and XFJ). However, the HI parameter and algal monosaccharide contents showed the same changing trends, and were positively correlated with $T_5$ at each of the reservoirs. Thus, the pyrolytic organic parameter and monosaccharide contents were more reliable for reconstructing of historical productivity in subtropical reservoirs.

In general, the corrected carbon isotopic composition could reflect the history of total productivity in one of the sediment cores. However, it is also affected by natural biogeochemical processes and anthropogenic activities. The pyrolytic organic parameter was observed to be specific to the type of NOM (e.g. algal NOM) and could help to distinguish the relative contribution of algae and higher plant to the NOM. By applying the molecular proxy of monosaccharides, the sources and detailed information of sedimentary organic matter can be provided. Therefore, the combination uses of these parameters are strongly recommended, which can help us to better understand the historical changes of past aquatic productivity and environment in the subtropical regions.

2. Second, the correlation between air-temperature changes in South China over 60 years and the trend in organic parameters of the three studied lakes is interesting. This part might be worth exploring in the future. Thus, a research outlook could be given by the authors.

**Response:**

We have added an outlook of research on the correlation of air temperature changes and organic parameters.

Elevated air-temperature could be the main driving factor in increasing productivity in Arctic lakes and some subtropical reservoirs. Sedimentary organic matter and their biomarker proxies of historical productivity are important to investigate the relationship between historical productivity and air-temperature variation. However, sedimentary organic matter comes from a variety of sources, including planktonic algae, terrestrial higher plants, zooplankton, organic detritus, black carbon, and so on. Moreover, most of organic matter is degraded during settling and post diagenesis.

Therefore, it is challenging to find the appropriate indicators for primary production in aquatic ecosystems. More works need to be done in this field on specific organic matter proxies of productivity. Meanwhile, multiple biomarker proxies are also needed to trace the source and type of NOM and to rule out the impact of human activities. Moreover, compound-specific isotope ratios, fractionations, and biodegradation products of biomarkers (e.g. neutral sugars, lipids) can provide more information of algal organic matter in aquatic ecosystems. Furthermore, the mechanism and modeling of relationships between air-temperature and algal organic matter parameters are worth to be exploited and established.

Specific comments/questions:

Line 24: The "single neutral carbohydrate" should be replaced by "monosaccharide", please correct it all throughout the text.

**Response:** We have changed "single neutral carbohydrate" to "monosaccharide" throughout the text according to your suggestion.

Line 47: The lowercase delta notation for isotopes should be italics. Please correct these all throughout the text.

**Response:** We have changed the lowercase delta to italics throughout the text according to your suggestion.

Line 94: The isotopic values are reported relative to the V-PDB Belemnite Standard, not just PDB.

**Response:** We have changed the "PDB" to "V-PDB Belemnite Standard" in the manuscript according to your suggestion.

Line 108: Is the "AC 5OW-X8" right? According to the Michael's paper in 2015, the cation resin should be "AG 50W-X8".

**Response:** We have changed the "AC 5OW-X8" to "AG 50W-X8" in the manuscript according
to your suggestion.

Line 243: "Fig. 3" should be "Fig. S3".

**Response:** We have changed the "Fig. 3" to "Fig. S4" in the manuscript according to your
suggestion.

Line 355: "Fig. S8" should be "Fig. S4".

**Response:** We have changed the "Fig. S8" to "Fig. S5" in the manuscript according to your
suggestion.

-------------------------------------------------------------------------------------------------------------------
**Anonymous Referee #2:**

This paper by Duan et al. compared organic matter characteristics ($\delta^{13}$C, C/N) and
monosaccharide distributions in sediment cores from three lakes with different depths (3 m, 17
m, and 36 m) and trophic states (mesotrophic vs. oligotrophic).

The neutral sugar data is nicely presented and discussed in the context of source and changes
in productivity and climate, making this a useful addition to the field. However, connections
between the carbon isotopic data and the neutral sugars are not clear in the text though
correlation between them is mentioned in the abstract and displayed in table S4. The
manuscript would benefit from expanding on the utility of combining these types of
measurements rather than discussing the data and their implications separately. After revising
this and the minor (but numerous) issues below, I would recommend the paper for publication
in Biogeosciences. In addition to these comments, the manuscript should be checked carefully
for small grammatical errors such as missing or incorrect articles and singular/plural
subject/verb issues.

**Response:**
We have added some paragraphs and cohesive discussions on the correlation analysis
and utility of combining these parameters and biomarkers. The differences and similarity of the corrected carbon isotopic composition, pyrolytic organic parameters and carbohydrate compositions have been presented and discussed. Please see the response to Reviewer 1.

The corrected carbon isotopic composition is a good indicator of aquatic productivity in some of aquatic ecosystems. Pyrolytic organic parameters could differentiate algal organic matter and terrestrial organic fractions. Monosaccharides compositions and some of the monosaccharide ratios are appropriate proxies for identifying the specific sources and types of NOM, which can be used to reflect the historical change of productivity in aquatic ecosystems.

**Minor comments:**

- pg 2, lines 44-48: Phytoplankton is plural so the verbs should be 'remove,' 'deplete,' and 'discriminate.'

**Response:** We have changed the "removes", "depletes", and "discriminates" to "remove", "deplete", and "discriminate" in the manuscript according to your suggestion.

Line 48 should be values.

**Response:** We have changed "value" to "values" in the manuscript according to your suggestion.

- pg 2, line 52: O'Reilly et al. (2005)

**Response:** We have added "et al." before "(2005)" in the manuscript according to your suggestion.

- pg 2, line 53: Verburg reference should be 2007

**Response:** We have changed "2006" to "2007" in the manuscript according to your suggestion.

- pg 2, line 55: It is not clear what 'it' in this sentence is referring to, please revise

**Response:** The "it" stands for "the $\delta^{13}C$ values in the reservoir sediment in the Pearl River Delta". We have revised it in the manuscript according to your suggestion.

- pg 3, line 58: Kirk et al. (2011)

**Response:** We have added "et al." before "(2011)" in the manuscript according to your
suggestion.

- pg 3, lines 66-67: Typo, add 'in' after 'help'; also 'Besides' is not correctly used here,
please revise

**Response:** We have added "in" after "help" in the manuscript according to your suggestion.
We have changed "Besides" to "Moreover" in the manuscript according to your suggestion.

- pg 4, line 94: is this actually V-PDB?

**Response:** Yes, it is "V-PDB". We have revised it in the manuscript according to your
suggestion.

- pg 4, line 94: from where is 'Product ID: GBW 04408' sourced?

**Response:** The product was purchased from National Research Center for Certified
Reference Materials (NRCRM), China. We have added the source of the product in the
manuscript.

- pg 4, lines 107-113: Michael et al., 2015 is not listed in the references

**Response:** The "Michael et al., 2015" is incorrect. We have revised it to "Philben et al., 2015"
in the citations throughout the manuscript.

- pg 6, line 192: Also not clear what 'it' refers to in this sentence, please clarify

**Response:** The "it" stands for "phytoplanktons". We have clarified "it" in the manuscript
according to your suggestion.

- pg 7, line 243: In this section (and in a few other places throughout the manuscript)
the monosaccharide names are strangely capitalized?

**Response:** monosaccharide names should not be capitalized. We have revised them throughout the manuscript according to your suggestion.

- pg 8, line 281: Hernes et al. (1996)

**Response:** We have added "et al." before "(1996)" in the manuscript according to your suggestion.

- pg 8, line 292: Keil et al. (1998)

**Response:** We have added "et al." before "(1998)" in the manuscript according to your suggestion.

- pg 9, line 308: Should be no 'et al.,' for Handa, 1969 reference

**Response:** We have deleted the "et al.," before "(1969)" in the manuscript according to your suggestion.

- pg 9, line 316: Another unclear 'it' usage, please revise

**Response:** The "it" stands for "the $k$ values of deoxy S/C5 in ZT, LA and XFJ sediments are low". We have clarified "it" in the manuscript according to your suggestion.

- pg 9, line 329: Gasse et al. should be 1991 as listed in the references

**Response:** We have revised "2001" to "1991" in the citation according to your suggestion.

- pg 9, lines 331-332: The use of 'algae-dominated' and then 'usually dominated in : : : algaes' is redundant. Additionally, the wording of 'dominated in' as a verb is grammatically incorrect (perhaps 'are usually dominant in'?) and 'algaes' is plural without the 's'

**Response:** We have deleted the "algae-dominated". We will change "dominated" to "dominant". We will change "algaes" to "algae" in the manuscript according to your suggestion.

- pg 9, line 332: should be Haug and Myklestad, 1976

**Response:** We have revised "Haug et al., 1976" to "Haug and Myklestad, 1976" in the
manuscript according to your suggestion.

- pg 9, line 334: typo ': : : the a: : :'; remove either 'the' or 'a'

**Response:** We have deleted "the" in the manuscript according to your suggestion.

- pg 10, line 340: the / between 'no/or' is not needed; alternatively the 'or' could be
removed ('no/weak correlations')

**Response:** We have revised "no/or correlations" to "no/weak correlation" in the manuscript
according to your suggestion.

- pg 10, line 367: this should be changed to 'neutral sugars : : : are'

**Response:** We have changed "is" to "are" in the manuscript according to your suggestion.

- pg 11, line 379: change 'are' to 'is'

**Response:** We have changed "are" to "is" in the manuscript according to your suggestion.

- pg 11, line 385: insert 'the' before 'last six decades'

**Response:** We have inserted "the" before "last six decades" in the manuscript according to
your suggestion.

- Figure 1: Is it possible to use the same scale for all three isotope profiles? Perhaps
with a range from -28 to -18 so that the reader can easily compare the three sites
visually

**Response:** Yes, it is. We have changed them to same scale for all three isotope profiles in the
manuscript according to your suggestion.

- Figure 2: The concentration range on the x-axis is quite large for the data, making it difficult to see variations with depth. Aside from the single outlier in the LA glucose profile, could these be changed to more appropriate ranges for the data?

**Response:** Yes, it is. We have changed these to more appropriate ranges for the data in the manuscript according to your suggestion.

- Borch et al. 1997, Gu et al. 2004, Kaiser and Benner 2000, Marchand et al. 2008,

Philben et al. 2015, Mopper et al. 1992, Ran et al. 2007, and Wakeham et al. 1997

are listed in the references but not cited in the text.

**Response:** The revisions have been made according to your suggestion:

1) We have deleted "Borch et al. 1997" in the reference    list.

2) "Gu and Schelske, 2004" should be "Gu et al., 2004", we have revised it in the citation throughout the manuscript.

3) Page 4, line 115: "Kaiser and Benner 2009" should be "Kaiser and Benner 2000" , we have revised it in the manuscript.

4) We have deleted the "Marchand et al. 2008" in the reference list.

5) The "Michael et al., 2015" should be "Philben et al., 2015", we have revised it in the citation throughout the manuscript.

6) We have deleted "Mopper et al. 1992" in the reference list.

7) We have deleted "Ran et al. 2007" in the reference    list.

8) We have deleted "Wakeham et al. 1997" in the reference    list.

--------------------------------------------------------------------------------------------------------------
**Anonymous Referee #3:**
Duan et al. obtained a very nice, enriched, dataset including neutral sugars and other
parameters in three subtropical reservoirs. Based on the concentrations and composition of the
neutral sugars, isotope values of TOC, and C/N ratios, they investigated source and diagenesis
pathways of sedimentary organic matter (SOM). They concluded that the dominant source of
SOM was phytoplankton in the ZT, LA and upper XFJ reservoirs, and there was not much
degradation of carbohydrates downward in the sediment cores. Also, there seems to be a nice
correlation between temperature and the levels of carbohydrates over the past 60 years. I think
this paper would be of interest to the community and worthy of being published,
(1) but I have issues with the way they presented, too broad and without a clear focus. The
authors discussed a lot of possible sources and phytoplankton among different reservoirs, but
they did not even mention why different patterns, ZT and LA vs. XFJ, were observed,. In
addition, some of the conclusions are very speculative. Overall, I do not feel this paper is ready
without a major revision.
**Reponses:**
We have made some revisions on the discussion in order to refine a clear focus:   The
combined uses of neutral sugars, carbon isotopic composition, and pyrolytic organic
parameters are recommended for reflecting the historical changes of productivity in subtropical
reservoirs. They can also be used for investigating climate change effects on algal productivity
in these reservoirs.
We also added detailed discussions on the causes and reasons for the possible sources
among different reservoirs. They are related not only to inputs of algae and plant, and bacteria,
but also to historical changes of hydrological conditions, nutrient level, anthropogenic activities,
and so on.
(2) The section of Materials and Methods needs more work. They need to include the
information about measuring the sedimentation rate and pyrolysis. I know they have these in
the Duan et al. 2015 paper, but these should be briefly described, since they use those data in
the Results section and you can't force the audience to read your other paper. It is unclear how
many cores they collected. In other words, how representative are these cores to the whole reservoirs. If these systems have been impacted by human activities, such as dredging,
sediments in these reservoirs can be very heterogeneous.

**Reponses:**
We have added the measurements for the sedimentation rate and pyrolysis according to
your suggestion.

We have sampled 2 or 3 cores for each reservoir, and each of the sediment cores were
collected in the center of the reservoir. Moreover, the reservoirs are mainly supplied by rainfall,
and are far away from the industrial center. The aquaculture is forbidden, and there are no
dredging activities in the investigated areas.

(3) A main issue with the manuscript is the lack of focus on the discussion. They talked about a
lot of difference topics, but it was written like a result section with titles like, "OM characteristics",
"Monosaccharide composition", "Source of neutral carbohydrates", and so on. In other words, it
reads more like a data report rather than a research paper.

**Reponses:**
The object of this study is to validate the combined uses of the carbon isotopic composition,
pyrolytic organic parameters, and neutral sugars as the potential proxies for historical changes
of productivity in subtropical reservoirs and their relationships with the climate changes in the
investigated areas.
The section of "OM characteristics" was written for the applicability of pyrolytic organic
parameters as algal proxies. Both "Monosaccharide composition" and "Source of neutral
carbohydrates" sections were compiled for the applicability of neutral sugars as algal proxies in
the investigated areas. We have made some revisions to emphasize a clear focus according to
your suggestion.

**Specific comments:**

Line 43: "offer"

**Reponses:** We have changed "offers" to "offer" in the manuscript according to your
suggestion.
Line 49: delete "and impacted"
**Reponses:** We have deleted the "and impacted" in the manuscript according to your
suggestion.
Line 54: any evidence about the Suess effect would be particularly stronger in the industrialized
areas such as Pearl River Delta? I would assume this should be about the same worldwide
considering the fast $CO_2$ mixing in the air.
**Reponses:** With rapid economic and industrial development in the Pearl River Delta, the local
lakes and reservoirs is more easily affected by the Suess effect due to the high-emssion of $CO_2$
even though the $CO_2$ mixing is fast in the air. However, there is no data for the Suess effect in
this area. Therefore, we have made some revisions on the description of the Suess effect in the
manuscript according to your suggestion.
Line 127: awkward wording, should be "productivity significantly contributed to dissolved
oxygen content"
**Reponses:** We have changed "the important role of oxygen content in the growth of
productivity" to "productivity significantly contributed to dissolved oxygen content" in the
manuscript according to your suggestion.
Line 130: nutrients levels are always higher in the deeper depth. What do you mean
by "be brought" to deeper depths"?
**Reponses:** During winter, the top layers of the lake have relative higher levels of productivity,
and the bottom layers have higher contents of nutrient. Moreover, the water column mixes from
top to bottom in the lake due to the decrease in temperature (so-called autumn overturn).
Therefore, the relative high contents of nutrients can be transported by the water flow to the
upper depths, resulting in the increase of nutrients and productivity in the entire water column.
We have made some revisions in the manuscript according to your suggestion.

Line 136: again, describe the pyrolysis

**Reponses:** We have added the instruction and description of pyrolysis in the manuscript
according to your suggestion.

Line 193-196: have to be careful about the C/N ratios. Decomposition of terrestrial organic
matter can decrease C/N ratios, not necessarily source related. This has been well
documented in composting studies. Also, the C/N ratios of 3 in the XFJ upper layers should be
interpreted in a more careful way. I don't think you can simply say "algal origin", because C/N
ratios fresh algae are typically about 6-7, and even pure bacterial biomass typically have C/N
ratios of 4. It is not very clear how you would get SOM with such low C/N ratios

**Reponses:** The very low C/N ratios are likely to be related to inorganic N in minerals. As the
TOC contents are quite low in XFJ, their inorganic N contents will affect the C/N ratios. We
have discussed this effect in the revised manuscript.

Line 200: the removal of $CH_4$ ($^{13}C$ light) should lead to the accumulation of $^{13}C$ heavy SOM

**Reponses:** We have changed "The removal of $^{12}CH_4$ by intensive methanogenesis also leads
to the accumulation of $^{13}C$-depleted OM" to "the removal of $CH_4$ ($^{13}C$ light) should lead to the
accumulation of $^{13}C$ heavy SOM" in the manuscript according to your suggestion.

Line 214-216: too speculative. The DO level you mentioned refers to the water, not sediment. I
think the major OM decomposition in these OM-enriched sediments is through anaerobic
pathway, unless you have DO profile data in sediment cores.

**Reponses:** We have deleted the speculative part and rewrite the paragraph in the manuscript
according to your suggestion.

Line 270: it's interesting to note the correlations between Zn and Cu and carbohydrates. I think more data analysis is needed, such as the contents of Zn and Cu in algae and how they trace metal got preserved, etc. It's not enough to simply have a correlation and then argue they were from phytoplankton. For example, it could have been sourced from industry contamination.

**Reponses:** We don't have Zn and Cu data in algae from the investigated areas. However, the Pb contents in the sediments of these reservoirs are very low, suggesting that there is no or little industry contamination in investigated areas.

Section 4.4. When the individual carbohydrates are normalized to TOC, I don't think there is much a decreasing trend at all (Table S2). In other words, carbohydrates simply are not good indicators of digenesis. This section should be strongly condensed.

**Reponses:** We have condensed the section 4.4 in the manuscript according to your suggestion. More investigations are needed to understand the fractions and degradation products of neutral carbohydrates.

Section 4.5. This section is interesting, but still at a speculative stage. Issues why we would expect carbohydrate increase, such as increased phytoplankton production or decomposition of SOM under warmer climate?

**Reponses:** We have observed significant correlations among $T_5$ temperature and a few of algal monosaccharides in the investigated reservoirs. Moreover, each of these monosaccharides is positively and significantly related to algal parameters (e.g. HI and S2). However, the diagenesis processes of neutral sugars and OM are estimated to be quite slow in the bottom sediments. Some fractions could be selectively preserved and remained unchanged during the post deposition. Therefore, the productivity proxies derived from some of the neutral sugars could be significantly related to the climate warming.

[revised manuscript text omitted]
  values and  neutral sugar data, show the same changing trends and are positively correlated with five $T_5$ at each of the three reservoirs. Thus, the HI parameters and monosaccharide contents are more reliable for reconstructing  historical productivity in subtropical reservoirs. Therefore, the combined use of these parameters and biomarkers are strongly recommended, which can help us to better understand the historical change of productivity and environments in the subtropical reservoirs.

It is challenging to find the appropriate indicators for primary production in aquatic ecosystems. More work need to be done  on specific organic matter for productivity proxy. Meanwhile, multiple biomarker proxies are also need in order to trace the source and type of biological productivity, and to rule out the impact of other human activities. Moreover, compound-specific isotope ratios, fractionation, and biodegradation products of biomarkers such as (e.g. neutral sugars, lipids) can provide more accurate and detailed information onf 
[revised manuscript text omitted]